# Forster Decomposition and Learning Halfspaces with Noise

**Ilias Diakonikolas**
University of Wisconsin-Madison
ilias@cs.wisc.edu

**Daniel M. Kane**
University of California, San Diego
dakane@cs.ucsd.edu

**Christos Tzamos**
University of Wisconsin-Madison
tzamos@wisc.edu

## Abstract

A Forster transform is an operation that turns a distribution into one with good anti-concentration properties. While a Forster transform does not always exist, we show that any distribution can be efficiently decomposed as a disjoint mixture of few distributions for which a Forster transform exists and can be computed efficiently. As the main application of this result, we obtain the first polynomial-time algorithm for distribution-independent PAC learning of halfspaces in the Massart noise model with strongly polynomial sample complexity, i.e., independent of the bit complexity of the examples. Previous algorithms for this learning problem incurred sample complexity scaling polynomially with the bit complexity, even though such a dependence is not information-theoretically necessary.

## 1  Introduction

The motivating application for this paper is the problem of (distribution-independent) PAC learning of halfspaces in the presence of label noise, and more specifically in the Massart (or bounded noise) model. Recent work [DGT19] obtained the first computationally efficient learning algorithm with non-trivial error guarantee for this problem. Interestingly, the sample complexity of the [DGT19] algorithm scales polynomially with the *bit complexity of the examples* (in addition, of course, to the dimension and the inverse of desired accuracy). This bit-complexity dependence in the sample complexity is an artifact of the algorithmic approach in [DGT19]. Information-theoretically, no such dependence is needed — alas, the standard VC-dimension-based sample upper bound [MN06] is non-constructive. Motivated by this qualitative gap in our understanding, here we develop a methodology that leads to a computationally efficient learning algorithm for Massart halfspaces (matching the error guarantee of [DGT19]) with "strongly polynomial" sample complexity, i.e., sample complexity completely independent of the bit complexity of the examples.

**Halfspaces and Efficient Learnability**   We study the binary classification setting, where the goal is to learn a Boolean function from random labeled examples with noisy labels. Our focus is on the problem of learning *halfspaces* in Valiant's PAC learning model [Val84] when the labels have been corrupted by *Massart noise* [MN06].

A halfspace is any function $h : \mathbb{R}^d \to \{\pm 1\}$ of the form $h(x) = \text{sign}(w \cdot x - \theta)$, where the vector $w \in \mathbb{R}^d$ is called the weight vector, $\theta \in \mathbb{R}$ is called the threshold, and $\text{sign} : \mathbb{R} \to \{\pm 1\}$ is defined by $\text{sign}(t) = 1$ if $t \geq 0$ and $\text{sign}(t) = -1$ otherwise. Halfspaces (or Linear Threshold Functions) are a central concept class in learning theory, starting with early work in the 1950s [Ros58, Nov62, MP68] and leading to fundamental and practically important techniques [Vap98, FS97]. Learning halfspaces

35th Conference on Neural Information Processing Systems (NeurIPS 2021).

is known to be efficiently solvable without noise (see, e.g., [MT94]) and computationally hard with adversarial (aka, agnostic) noise [GR06, FGKP06, Dan16]. The Massart model is a natural compromise, in the sense that it is a realistic noise model that may allow for efficient algorithms without distributional assumptions. An essentially equivalent formulation of this model was already defined in the 80s by Sloan and Rivest [Slo88, Slo92, RS94, Slo96], and a very similar definition had been considered even earlier by Vapnik [Vap82].

**Definition 1.1** (Massart Noise). Let $\mathcal{C}$ be a class of Boolean functions over $X = \mathbb{R}^d$, $\mathcal{D}_X$ be a distribution over $X$, and $0 \leq \eta < 1/2$. Let $f$ be an unknown target function in $\mathcal{C}$. A *noisy example oracle*, $\mathrm{EX}^{\mathrm{Mas}}(f, \mathcal{D}_X, \eta)$, works as follows: Each time $\mathrm{EX}^{\mathrm{Mas}}(f, \mathcal{D}_X, \eta)$ is invoked, it returns a labeled example $(x, y)$, where $x \sim \mathcal{D}_X$, $y = f(x)$ with probability $1 - \eta(x)$ and $y = -f(x)$ with probability $\eta(x)$, for an *unknown* parameter $\eta(x) \leq \eta$. Let $\mathcal{D}$ denote the joint distribution on $(x, y)$ generated by the above oracle. The learner is given i.i.d. samples from $\mathcal{D}$ and wants to output a hypothesis $h$ such that with high probability the error $\mathbf{Pr}_{(x,y)\sim\mathcal{D}}[h(x) \neq y]$ is small.

In recent years, the algorithmic task of learning halfspaces in the Massart model has attracted significant attention in the learning theory community. In the *distribution-specific* PAC model, where structural assumptions are imposed on the marginal distribution $\mathcal{D}_X$, the work of [ABHU15] initiated the study of learning homogeneous halfspaces with Massart noise. Since this work, a sequence of papers [ABHZ16, YZ17, ZLC17, BZ17, DKTZ20, ZSA20, ZL21] developed efficient algorithms for this problem, achieving error $\mathrm{OPT} + \epsilon$, for various classes of structured distributions.

In the *distribution-independent* setting, where no assumptions are made on the distribution $\mathcal{D}_X$, progress has been slow. The existence of an efficient PAC learning algorithm for Massart halfspaces had been a long-standing open question posed in a number of works [Slo88, Coh97, Blu03], with no algorithmic progress until recently. Recent work [DGT19] gave the first polynomial-time algorithm with non-trivial error guarantees for this problem, specifically achieving error $\eta + \epsilon$, where $\eta$ is the upper bound on the noise rate. Since the dissemination of [DGT19], a number of works have studied the complexity of learning halfspaces and other concept classes in the presence of Massart noise. [CKMY20] gave a proper learning algorithm for Massart halfspaces matching the error guarantee of [DGT19]. On the lower bound side, hardness results were established for both exact [CKMY20] and approximate learning [DK20]. Specifically, [DK20] gave a Statistical Query (SQ) lower bound ruling out efficient SQ learning algorithms achieving *any* polynomial relative approximation for Massart halfspaces. Interestingly, the [DGT19] approach also motivated the design of the first boosting algorithm for Massart PAC learning [DIK+21]. The boosting algorithm of [DIK+21] efficiently achieves error $\eta + \epsilon$, for any concept class, starting from a black-box Massart weak learner.

To motivate our results, we state the guarantees of the [DGT19] algorithm in more detail. (The proper algorithm of [CKMY20] builds on the same ideas and has qualitatively similar guarantees.) Let $\mathcal{D}_X \subset \mathbb{R}^d$ be the marginal distribution on the examples and $b$ be an upper bound on the bit complexity of points $x$ in the support of $\mathcal{D}_X$. Then, the algorithm of [DGT19] requires $n = \mathrm{poly}(d, b, 1/\epsilon)$ labeled examples, runs in time $\mathrm{poly}(n, b)$ and achieves misclassification error $\eta + \epsilon$. The dependence on $b$ in the runtime is likely to be inherent. (Learning halfspaces *without* noise amounts to solving a general linear program (LP); removing the $b$ dependence from the runtime would yield a strongly polynomial algorithm for general LP.) On the other hand, there is no a priori reason to believe that the $\mathrm{poly}(b)$ dependence is needed in the sample complexity. In fact, it is known [MN06] that $\mathrm{poly}(d/\epsilon)$ samples *information-theoretically suffice* to achieve optimal misclassification error. Of course, this sample complexity bound is non-constructive, in the sense that the sample upper bound argument does not yield a sub-exponential time learning algorithm. The above discussion motivates the following natural question: *Is there an efficient learning algorithm for Massart halfspaces using only $\mathrm{poly}(d/\epsilon)$ samples?* The main result of this paper provides an affirmative answer to this question.

## 1.1 Our Results and Techniques

The main learning theory result of this paper is the following.

**Theorem 1.2** (Main Learning Result). *There is an algorithm that for all $0 < \eta < 1/2$, on input a set of $n = \mathrm{poly}(d, 1/\epsilon)$ i.i.d. examples from a distribution $\mathcal{D} = \mathrm{EX}^{\mathrm{Mas}}(f, \mathcal{D}_X, \eta)$ on $\mathbb{R}^{d+1}$, where $f$ is an unknown halfspace on $\mathbb{R}^d$, it runs in $\mathrm{poly}(n, b, 1/\epsilon)$ time, where $b$ is an upper bound on the bit complexity of the examples, and outputs a hypothesis $h$ that with high probability satisfies $\mathbf{Pr}_{(x,y)\sim\mathcal{D}}[h(x) \neq y] \leq \eta + \epsilon$.*

**Brief Overview of [DGT19] Algorithm**    To explain the source of our qualitative improvement, we start with a high-level description of the previous algorithm from [DGT19]. At a high-level, this algorithm works in two steps: First, one designs an efficient learner for the special case where the target halfspace has some non-trivial *anti-concentration* (aka "large" margin). Then, one develops an efficient reduction of the general (no margin) case to the large-margin case. Formally speaking, such a reduction is not entirely "black-box"; this description is for the purpose of intuition.

First, we note that without loss of generality the target halfspace is homogeneous (since we are working in the distribution-independent setting). The aforementioned reduction, used in [DGT19], relies on a method from [BFKV96] (refined in [DV04]). The idea is to slightly modify the distribution on the unlabeled points to guarantee a (weak) margin property. After this modification, there exists an explicit margin parameter $\sigma = \Omega(1/\mathrm{poly}(d, b))$, such that any hyperplane through the origin has a non-trivial mass of the distribution at distance at least $\sigma$ standard deviations from it. If $\sigma$ is a bound on the margin, the algorithm developed in step one has sample complexity (and running time) $\mathrm{poly}(d, 1/\sigma, 1/\epsilon)$. This is the source of the "$b$-dependence" in the sample complexity of the [DGT19] algorithm (recalling that $\sigma = \Omega(1/\mathrm{poly}(d, b))$).

As we will explain below, the approach of [BFKV96, DV04] inherently leads to a "$b$-dependence" in the margin parameter $\sigma$. At a very high-level, the key to our improvement is to develop a new efficient preprocessing routine that achieves $\sigma = \Omega(1/\mathrm{poly}(d))$. This leads to the desired strongly polynomial sample complexity.

**Preprocessing Requirements**    Before we can get into the details of these preprocessing algorithms, we first need to specify the anti-concentration property that we need to guarantee. Essentially, our algorithms need there to be a decent fraction of points that are reasonably far from the defining hyperplane. More specifically, if the true separating hyperplane is defined by a linear function $L(x) = w^* \cdot x$ (for some weight vector with $\|w^*\|_2 = 1$), we need that a non-trivial fraction of points $x$ (say, a $1/\mathrm{poly}(d)$-fraction) should have $|L(x)|$ be a non-trivial fraction of $\mathbf{E}[L^2(x)]^{1/2}$.

One might ask how we can hope to achieve such a guarantee without knowing the true separator $L(x)$ ahead of time. This can be guaranteed if, for example, the points are in *radial isotropic position*. In particular, this means that for every point $x$ in the support of our distribution it holds that $\|x\|_2 = 1$ and that $\mathbf{E}[xx^T] = (1/d)\, I_d$, where $I_d$ is the $d \times d$ identity matrix. The latter condition implies that $\mathbf{E}[L^2(x)] = 1/d$. Combining this with the fact that $|L(x)| \leq 1$ for all points $x$ in the support, it is not hard to see that with probability at least $1/d$ we have that $L^2(x)$ is at least $1/d$ — implying an appropriate anti-concentration bound. In fact, it will suffice to have our point set be in *approximate* radial isotropic position, allowing $\mathbf{E}[xx^T]$ to merely be proportional to $(1/d)\, I_d$ and allowing $\|x\|_2$ to be more than one, as long as it satisfies some polynomial upper bound. We note that the size of this upper bound will affect the quality of the anti-concentration result, and hence the performance of the remainder of the algorithm.

Unfortunately, not all point sets are in radial isotropic position. However, there is a natural way to try to fix this issue. It is not hard to see that our original halfspace learning problem is invariant under linear transformations, and it is a standard result that (unless our support lies in a proper subspace) there always exists a linear transformation that can be applied to ensure that $\mathbf{E}[xx^T] = (1/d)\, I_d$. However, after applying this transformation, we may still have points whose norms are too large. Essentially, we need to ensure that no point is too large in terms of *Mahalanobis distance*. Namely, that if $\Sigma = \mathbf{E}[xx^T]$, we want to ensure that $|x^T \Sigma^{-1} x|$ is never too large. Unfortunately, in the data set we are given, this might still not be the case.

**Preprocessing Routine of [BFKV96, DV04]**    The key idea in [BFKV96] and [DV04] is to find a core set $S$ of sample points such that the Mahalanobis distance of the points in $S$ (with respect to the second-moment matrix of $S$) is not too large. This will correspond to a reasonable sized sub-distribution of our original data distribution on which our desired anti-concentration bounds hold, allowing our learning algorithm to learn the classifier at least on this subset. In [BFKV96], it is shown that this can be achieved by a simple iterative approach, where points with too-large Mahalanobis norm are repeatedly thrown away. This step needs to be performed in several stages, as throwing away points will alter the second-moment matrix, and thus change the norm in question. Interestingly, [BFKV96] show that the number of iterations required by this procedure can be bounded in terms of the numerical complexity of the points involved.

Unfortunately, in order to avoid throwing away too many of the original samples, the procedure in [BFKV96] needs to sacrifice the quality of the resulting anti-concentration bound. This reduced quality will result in an increased sample complexity of the resulting algorithm, in particular causing it to scale polynomially with the bit complexity of the samples. The subsequent work [DV04] makes some quantitative improvements to the "outlier removal" procedure, however the final anti-concentration quality still has polynomial dependence on the bit complexity of the inputs, which is then passed on to the sample complexity of the resulting algorithm. What is worse is that [DV04] prove a *lower bound* showing that *any* outlier removal algorithm must have a similar dependence on the bit complexity.

**Our Approach**  The aforementioned lower bound of [DV04] shows that no combination of point removal and linear transformation can put the input points into approximate radial isotropic position without losing polynomial factors of the bit complexity in either the quality or the fraction of remaining points. However, there is another operation that we can apply to the data without affecting the underlying learning problem.

In particular, since we are dealing with homogeneous halfspaces (without loss of generality), the problem will be unaffected by replacing a sample point $x$ with the point $\lambda\, x$ for any scaling factor $\lambda > 0$ (since $\text{sign}(L(x)) = \text{sign}(L(\lambda\, x))$ no matter what $L$ is). This gives us another tool to leverage in our preprocessing step.

In particular, by applying an appropriate linear combination to our points, we can ensure that they are in isotropic position (i.e., having second-moment matrix $(1/d)\, I_d$). Similarly, by rescaling individual points, we can ensure that our points are in radial position (i.e., that $\|x\|_2 = 1$ for all $x$). The question we need to ask is whether by applying some combination of these two operations, we can ensure that *both* of these conditions hold simultaneously. In other words, we would like to find an invertible linear transformation $A$ such that after replacing each point $x$ by the point $Ax/\|Ax\|_2$ (in order to make it unit-norm), the resulting points are in (approximate) isotropic position.

The problem of finding such transformations was studied in early work by Forster [For02] (see also [Bar98]) who showed that it is possible to achieve *under certain assumptions*, including for example the natural setting where the points are drawn from a continuous distribution. Unfortunately, there are cases where appropriate linear transformations $A$ do not exist. In particular, if there was some $d/3$-dimensional subspace that contained half of the points, then after applying *any* such transformation to our dataset, this will still be the case, and thus there will be a $d/3$-dimensional subspace over which the trace of the covariance matrix is at least $1/2$. Interestingly, in a refinement of Forster's work, the recent work [HKLM20] showed that this is the only thing that can go wrong. That is, a matrix $A$ exists unless there is a $k$-dimensional subspace containing more than a $k/d$-fraction of the points.

Suppose that we end-up in the latter case. Then, by restricting our attention to only the points of this subspace (and a subspace of that, if necessary), we can always find a relatively large subset of our initial dataset so that after applying a combination of linear transformations and pointwise-rescaling, they can be put into radial isotropic position.

Of course, to take advantage of such a transformation, one must be able to find it efficiently. While prior work [HM13, AKS20] has obtained algorithmic results for this problem, none appear to apply in quite the generality that we require. Our main algorithmic result is that such transformations exist and can be efficiently (approximately) computed. We start with the following simple definition:

**Definition 1.3.**  Given an inner product space $V$ and an invertible linear transformation $A : V \to V$, we define the mapping $f_A : (V \backslash \{0\}) \to (V \backslash \{0\})$ by $f_A(x) = Ax/\|Ax\|_2$.

We can now state our main algorithmic result (see Theorem 3.4 for a more detailed statement):

**Theorem 1.4** (Algorithmic Generalized Forster Transform). *There exists an algorithm that, given a set $S$ of $n$ points in $\mathbb{Z}^d \backslash \{0\}$ of bit complexity at most $b$ and $\delta > 0$, runs in $\text{poly}(n, d, b, \log(1/\delta))$ time and returns a subspace $V$ of $\mathbb{R}^d$ containing at least a $\dim(V)/d$-fraction of the points in $S$ and a linear transformation $A : V \to V$ such that $\frac{1}{|S \cap V|} \sum_{x \in S \cap V} f_A(x) f_A(x)^T = (1/\dim(V))\, I_V + O(\delta)$ , where the error is in spectral norm.*

By applying Theorem 1.4 iteratively to the points of $S \setminus (S \cap V)$, we obtain a decomposition of $S$ into not too many subsets $T$, so that each $T$ has a Forster transform over the subspace which it spans.

## 1.2 Preliminaries

For $n \in \mathbb{Z}_+$, we denote $[n] \stackrel{\text{def}}{=} \{1, \ldots, n\}$. We write $E \gtrsim F$ to denote that $E \geq c\,F$, where $c > 0$ is a sufficiently large universal constant. For $V \subseteq \mathbb{R}^d$ and $f : \mathbb{R}^d \to \mathbb{R}$, we use $\mathbf{1}_V(f)$ for the characteristic function of $f$ on $V$, i.e., $\mathbf{1}_V(f) : V \to \{0, 1\}$ and $\mathbf{1}_V(f)(x) = 1$ iff $f(x) \neq 0$, $x \in V$.

For a vector $x \in \mathbb{R}^d$, and $i \in [d]$, $x_i$ denotes the $i$-th coordinate of $x$, and $\|x\|_2 \stackrel{\text{def}}{=} (\sum_{i=1}^d x_i^2)^{1/2}$ denotes the $\ell_2$-norm of $x$. We will use $x \cdot y$ for the inner product between $x, y \in \mathbb{R}^d$. For a (linear) subspace $V \subset \mathbb{R}^d$, we use $\dim(V)$ to denote its dimension. For a set $S \subset \mathbb{R}^d$, $\text{span}(S)$ will denote its linear span. For a matrix $A \in \mathbb{R}^{d \times d}$, we use $\|A\|_2$ for its spectral norm and $\text{tr}(A)$ for its trace. For $A, B \in \mathbb{R}^{d \times d}$ we use $A \succeq B$ for the Loewner order, indicating that $A - B$ is positive semidefinite (PSD). We denote by $I_d$ the $d \times d$ identity matrix and by $I_V$ the identity matrix on subspace $V$.

We use $\mathbf{E}[X]$ for the expectation of $X$ and $\mathbf{Pr}[\mathcal{E}]$ for the probability of event $\mathcal{E}$. For a finite set $S$, we will use $x \sim_u S$ to denote that $x$ is drawn uniformly at random from $S$.

## 2 Algorithmic Forster Decomposition: Proof of Theorem 1.4

Given a distribution $X$ on $\mathbb{R}^d$, our goal is to transform $X$ so that the transformed distribution has good anti-concentration properties. Specifically, we would like it to be the case that for any direction $v$, there is a non-trivial probability that $|v \cdot X|^2$ is at least a constant multiple of $\mathbf{E}[|v \cdot X|^2]$. It is easy to see that this condition can be achieved as long as no particular value in the support of $X$ contributes too much to $\mathbf{E}[|v \cdot X|^2]$. In particular, it suffices that there exists some constant $B > 0$ such that $|v \cdot x|^2 < B\,\mathbf{E}[|v \cdot X|^2]$ for all vectors $v$ and all $x$ in the support of $X$. If this holds, it is easy to see that with at least $\Omega(1/B)$ probability a randomly chosen $x \sim X$ satisfies $|v \cdot x|^2 \geq \mathbf{E}[|v \cdot X|^2]/2$.

Unfortunately, a given distribution $X$ may not have this desired property. However, it seems in principle possible that one can modify $X$ so that it satisfies this property. In particular, for any given weighting function $c : \mathbb{R}^d \to \mathbb{R}_+$, we can replace the distribution $x \sim X$ with the distribution $c(x)x$ without affecting our linear classifier. Intuitively, by scaling down the outliers, we might hope that this kind of scaling would have the desired properties. This naturally leads us to a number of questions to be addressed:

1. How do we know that such a weighting function $c$ exists?
2. If such a function exists, (how) can we efficiently compute a function $c$, even for a discrete distribution $X$?
3. If $X$ has continuous support, how can we find a function $c$ that works for $X$, given access to a small set of i.i.d. samples?

To address these questions, it will be useful to understand the second moment (autocorrelation) matrix of the transformed random variable $c(X)X$, i.e., $\Sigma = \mathbf{E}[(c(X)X)(c(X)X)^T]$. Observe that our desired condition boils down to $|v \cdot c(x)x|^2 \leq Bv \cdot \Sigma v$, or equivalently

$$c(x) \leq B \inf_{v \neq 0} \sqrt{(v^T \Sigma v)/|v \cdot x|^2} \,, \tag{1}$$

for all points $x$ in the support of $X$. We note that this setup forces us to strike a balance between $c(x)$ being large and $c(x)$ being small. On the one hand, if $c(x)$ is too large, it will violate Equation (1). On the other hand, if $c(x)$ is too small, the expectation of $c(X)^2 XX^T$ will fail to add up to $\Sigma$. However, it is easy to see that making $\Sigma$ larger is never to our detriment; that is, given $\Sigma$, we might as well take $c(x)$ so that equality holds in Equation (1).

An additional technical difficulty here is related to the infimum term in Equation (1). This issue is somewhat simplified by making a change of variables so that the transformed autocorrelation matrix becomes equal to the identity, i.e., $\Sigma = I$. In this case, Equation (1) reduces to the condition $c(x) \leq B/\|x\|_2$, for $x$ in the support of $X$. Changing variables back, we can see that the original equation is equivalent to $c(x) \leq B/\|\Sigma^{-1/2}x\|_2$; however, making the change of variables explicitly will make it easier to relate this problem to existing work on Forster's theorem [For02].

If we find a linear transformation $A$ such that the matrix $\Sigma_A := \mathbf{E}[f_A(X)f_A(X)^T]$ satisfies $\Sigma_A \succeq (1/B)\,I$, then since each value of $f_A(X)$ is a unit vector, the distribution $f_A(X)$ will satisfy

our anti-concentration condition. Also observe that since $\mathrm{tr}(\Sigma_A) = 1$, we cannot expect the parameter $B > 0$ to be smaller than $\dim(V)$. Moreover, even this may not be possible in general. In particular, if a large fraction of the points in the support of $X$ lie on some proper subspace $W$, most of the mass of $f_A(X)$ will lie in the subspace $AW$. Therefore, the trace of $\Sigma_A$ along this subspace will be more than $\dim(W)/\dim(V)$, forcing some of the other eigenvalues to be smaller than $1/\dim(V)$.

Interestingly, Forster [For02] showed that unless many points of $X$ have these kinds of linear dependencies, then a linear transformation $A$ with the desired properties always exists. This condition was refined in a recent work [HKLM20] who proved the following existence theorem.

**Theorem 2.1** (Generalized Forster Transform). *Let $X$ be a distribution with finite support on an inner product space $V$. Then, unless there is a proper subspace $W$ of $V$ so that $\mathbf{Pr}[X \in W] \geq \dim(W)/\dim(V)$, there exists an invertible linear transformation $A : V \to V$ such that $\Sigma_V = (1/\dim(V))\,I$.*

Theorem 1.4 is an algorithmic version of the above theorem. The proof has two main ingredients. We start by handling the case of rescaling points rather than finding a linear transformation. It turns out that handling this case is quite simple, as shown in the following result.

**Proposition 2.2.** *There is an algorithm that, given a set $S$ of $n$ points in $\mathbb{Z}^d$ each of bit complexity at most $b$, lying in a subspace $V \subseteq \mathbb{R}^d$, and a parameter $\delta > 0$, runs in $\mathrm{poly}(n, d, b, \log(1/\delta))$ time and, unless there is a proper subspace $W \subset V$ containing at least a $\dim(W)/\dim(V)$-fraction of the points in $S$, returns a weight function $c : S \to \mathbb{R}^+$ such that for every $x \in S$ we have that*

$$c^2(x)xx^T \preceq \frac{\dim(V)+\delta}{n} \sum_{y \in S} c^2(y)yy^T \ . \tag{2}$$

*Moreover, the function $c^2$ takes integral values of bit complexity $\mathrm{poly}(n, d, b, \log(1/\delta))$.*

*Proof.* We start by showing that such a weight function $c$ exists. By Theorem 2.1, under the given condition on subspaces, there must exist an invertible linear transformation $A : V \to V$ such that $(1/n) \sum_{y \in S} f_A(y)f_A(y)^T = (1/\dim(V))\,I_V$. This means that for any $w \in V$ and $x \in S$, we have that $|w \cdot (Ax/\|Ax\|_2)|^2 \leq (\dim(V)/n) \sum_{y \in S} |w \cdot (Ay/\|Ay\|_2)|^2$. Rearranging, this implies that

$$|A^T w \cdot (x/\|Ax\|_2)|^2 \leq (\dim(V)/n) \sum_{y \in S} |A^T w \cdot (y/\|Ay\|_2)|^2 \ .$$

Thus, letting $c(x) = 1/\|Ax\|_2$ causes our desired Equation (2) to hold for all vectors $A^T w$. Since $A$ is invertible, this covers all vectors in $V$. Moreover, since all points in $S$ lie in $V$, that is sufficient to check in order to ensure that we satisfy Equation (2) in general.

We will show that we can efficiently compute values $c(x) > 0$ for each $x \in S$, such that Equation (2) holds. We note that this is just a semidefinite program (SDP) in the variables $c^2(x)$. The constraints can be written as: $c^2(x)xx^T \preceq (\dim(V)/n) \sum_{y \in S} c^2(y)yy^T$, for all $x \in S$, and the positivity constraint can be written as $c(x)^2 \geq 1$ for all $x \in S$. It remains to argue that the above SDP can be solved efficiently in time $\mathrm{poly}(n, d, b, \log(1/\delta))$, after relaxing the constraints by $\delta$ as in the theorem statement, via the Ellipsoid algorithm. This argument is somewhat technical and is deferred to the supplementary material. $\qquad\square$

It remains to handle the case when a proper subspace $W$ exists. We show that one can identify such a subspace efficiently.

**Proposition 2.3.** *There is a polynomial-time algorithm that, given a set $S$ of $n$ points in $\mathbb{Z}^d$ of bit complexity $b$ all lying in a subspace $V \subseteq \mathbb{R}^d$, determines whether or not there exists a proper subspace $W \subset V$ containing at least a $\dim(W)/\dim(V)$-fraction of the points in $S$, and if so returns such a subspace.*

*Proof.* Let $S = \{x^{(i)}\}_{i=1}^n \subset \mathbb{R}^d$, $V = \mathrm{span}(S)$, and $k = \dim(V)$. We will first show that if $n$ is a multiple of $k$, we can efficiently find a "heavy" subspace $W$ of dimension $\kappa = \dim(W)$, i.e., one with at least $\frac{n}{k}\kappa + 1$ points, if one exists. To achieve this, we set up the following feasibility linear program (LP), with a variable $v_i \in [0,1]$ for every point $x^{(i)}$. We require that for any subset $S' \subseteq S$ of $k$ linearly independent vectors the following linear inequality is satisfied

$$\sum_{i=1}^n v_i \geq \frac{n}{k} \sum_{x_i \in S'} v_i + 1 \ . \tag{3}$$

**Efficient Computation**   We note that even though there are exponentially many constraints, the above LP can be solved efficiently via the Ellipsoid algorithm. To show this, we provide a separation oracle that given any guess $v \in [0,1]^n$ efficiently identifies a violating constraint. In more detail, given any vector $v$, the linear independent basis $\mathcal{B}$ that maximizes the RHS of (3) can be efficiently computed by a greedy algorithm. Starting from the empty set, we repeatedly add one point at a time, at each step choosing a point $x^{(i)}$ of maximum $v_i$, among the elements whose addition would preserve the independence of the augmented set. The greedy algorithm correctly identifies a basis of maximum weight, as the family of linearly independent subsets of points forms a matroid.

**Feasibility**   We now show that the above LP will be feasible if and only if there exists a heavy subspace $W$. We start with the forward direction, i.e., assume the existence of a heavy subspace $W$. In this case, setting $v_i = 1$ if $x^{(i)} \in W$ and $v_i = 0$ otherwise, we can see that all constraints of the LP are satisfied:

- The LHS of (3) is always at least $\frac{n}{k}\kappa + 1$, since there at least these many points on the subspace $W$.

- The RHS of (3) is at most $\frac{n}{k}\kappa + 1$, as one can pick at most $\kappa$ points $x^{(i)}$ with corresponding values 1.

For the reverse direction, if the LP is feasible and we can find a feasible vector $v$, we can efficiently identify a heavy subspace $W$. To do this we proceed as follows: Assume w.l.o.g. that $v_1 \geq v_2 \geq \ldots \geq v_n$. Run the greedy algorithm described above to find the basis $\mathcal{B}$ with points $x^{(i_1)}, x^{(i_2)}, \ldots, x^{(i_d)}$, where $1 = i_1 < i_2 < \ldots < i_k$, that maximizes the RHS of (3) . Then, we find some $\kappa$ such that $i_{\kappa+1} > \frac{n}{k}\kappa + 1$. As we will soon show, such a $\kappa$ must always exist. Having such a $\kappa$ means that the first $\frac{n}{k}\kappa + 1$ points lie in a $\kappa$-dimensional subspace $W$, since otherwise the greedy algorithm would have picked the $(\kappa + 1)$-th point in the basis earlier in the sequence.

We now argue by contradiction that some $\kappa \in [0, k-1]$ such that $i_{\kappa+1} > \frac{n}{k}\kappa + 1$ must always exist. Indeed, suppose that for all $\kappa$, $i_{\kappa+1} \leq \frac{n}{k}\kappa + 1$. Then, we have that

$$v_{i_{\kappa+1}} \geq v_{\frac{n}{k}\kappa+1} \geq \frac{\sum_{j=1}^{n/k} v_{\frac{n}{k}\kappa+j}}{n/k} \ .$$

Summing the above, over all $\kappa \in [0, k-1]$, we get that

$$\sum_{x_i \in B} v_i \geq \frac{k}{n} \sum_{i=1}^{n} v_i \ ,$$

which leads to the desired contradiction contradiction, as this implies that $v$ is infeasible.

In summary, we have shown that when $n$ is a multiple of $k$, we can find a "heavy" subspace $W$ of dimension $\kappa = \dim(W)$ with at least $\frac{n}{k}\kappa + 1$ points. This is off-by-one by the guarantee we were hoping for, which was to find a subspace with at least $\frac{n}{k}\kappa$ points. We can address this issue by running our algorithm on a modified point-set after replacing one point outside the subspace of interest with one inside to increase the number of inliers by 1. Even though we do not know which pair of points to change, we can run the algorithm for all pairs of points until a solution is found. We can also handle the general case where $n$ is not a multiple of $k$, by making $k$ copies of every point and running the algorithm above. $\qquad\square$

We are now ready to prove Theorem 1.4.

*Proof of Theorem 1.4.*   The algorithm begins by iteratively applying the algorithm from Proposition 2.3 until it finds a subspace $V$ containing at least a $k/d$-fraction of the points in $S$, such that no proper subspace $W \subset V$ contains at least a $\dim(W)/\dim(V)$-fraction of the points in $S \cap V$.

We then apply the scaling algorithm of Proposition 2.2 to find a weight function $c : S \cap V \to \mathbb{R}^+$, and consider the matrix

$$A = \left[ \frac{1}{|S \cap V|} \sum_{x \in S \cap V} c(x)^2 x x^T \right]^{-1/2} \ .$$

We note that Equation (2) now reduces to the following

$$c^2(x)|w \cdot x|^2 \leq (\dim(V) + \delta)\|A^{-1}w\|_2^2 ,$$

for all vectors $w$. Setting $w = A^2x$, we obtain

$$c^2(x)\|Ax\|_2^4 \leq (\dim(V) + \delta)\|Ax\|_2^2 ,$$

which gives

$$c^2(x) \leq \frac{\dim(V) + \delta}{\|Ax\|_2^2}.$$

On the other hand, we have that

$$I_V = \frac{1}{|S \cap V|} \sum_{x \in S \cap V} c(x)^2 (Ax)(Ax)^T \preceq \frac{\dim(V) + \delta}{|S \cap V|} \sum_{x \in S \cap V} f_A(x)f_A(x)^T.$$

By the above inequality, it follows that all eigenvalues $\lambda_1, \ldots, \lambda_{\dim(V)}$ of the matrix $\frac{1}{|S \cap V|} \sum_{x \in S \cap V} f_A(x)f_A(x)^T$ are at least $\frac{1}{\dim(V)+\delta}$. However, since $\frac{1}{|S \cap V|} \sum_{x \in S \cap V} f_A(x)f_A(x)^T$ has trace 1, we also have that $\sum_{i=1}^{\dim(V)} \lambda_i = 1$. This means that the maximum eigenvalue of this matrix is at most $\frac{1+\delta}{\dim(V)+\delta}$. Therefore, $\frac{1}{|S \cap V|} \sum_{x \in S \cap V} f_A(x)f_A(x)^T$ is within $O(\delta)$ of $(1/\dim(V)) I_V$ in spectral norm. This completes the proof of Theorem 1.4. □

The final ingredient that will be important for us is showing that if we can find a subspace $V$ and linear transformation $A$ (as specified in the statement of Theorem 1.4) that works for a sufficiently large set of i.i.d. samples from the distribution $X$, then the same transform will work nearly as well for $X$. This is established in the following proposition.

**Proposition 2.4.** *Let $X$ be any distribution on $\mathbb{R}^d \setminus \{0\}$ and $S$ be a multiset of $n \gtrsim d^2/\epsilon^2$ i.i.d. samples from $X$. Then with high probability over the choice of $S$ the following holds: For every subspace $V$ of $\mathbb{R}^d$, every invertible linear transformation $A : V \to V$, and any unit vector $w \in V$, we have that:*

1. *$|S \cap V|/|S| = \mathbf{Pr}[X \in V] + O(\epsilon)$.*

2. *$(1/|S|) \sum_{x \in S \cap V} |w \cdot f_A(x)|^2 = \mathbf{E}[\mathbf{1}_V(X)|w \cdot f_A(X)|^2] + O(\epsilon)$.*

*Proof.* The proof is by a simple application of the VC inequality (see, e.g., [DL01]), stated below.

**Fact 2.5** (VC Inequality). *Let $\mathcal{F}$ be a class of Boolean functions with finite VC dimension $\mathrm{VC}(\mathcal{F})$ and let a probability distribution $D$ over the domain of these functions. For a set $S$ of $n$ independent samples from $D$, we have that*

$$\sup_{f \in \mathcal{F}} |\mathbf{Pr}_{X \sim S}[f(X)] - \mathbf{Pr}_{X \sim D}[f(X)]| \lesssim \sqrt{\frac{\mathrm{VC}(\mathcal{F})}{n}} + \sqrt{\frac{\log(1/\tau)}{n}} ,$$

*with probability at least $1 - \tau$.*

Item 1 follows directly by noting that the set of vector subspaces of $\mathbb{R}^d$ has VC-dimension $d$. Item 2 follows from the fact that the collection of sets

$$F_{V,A,w,t} = \{x \in V : |w \cdot f_A(x)|^2 \geq t\}$$

has VC-dimension $O(d^2)$. This holds for the following reason: A set of this form is the intersection of the subspace $V$ (which comes from a class of VC-dimension at most $d$), with the set of points $x$ such that the quadratic polynomial $(w \cdot (Ax))^2 - t\|Ax\|_2^2$ is non-negative. Recall that the space of degree-2 threshold functions is a class of VC-dimension $O(d^2)$. Therefore, by the VC inequality, with high probability, for each such $F$ we have that the fraction of $S$ in $F$ is within $O(\epsilon)$ of the probability that a random element of $X$ lies in $F$. The claim now follows by noting that for a distribution $Y$ (either $X$ or the uniform distribution over $S$) we have that

$$\mathbf{E}[\mathbf{1}_V(Y)|w \cdot f_A(Y)|^2] = \int_{t=0}^1 \mathbf{Pr}[Y \in F_{V,A,w,t}]dt .$$

This completes the proof of Proposition 2.4. □

# 3 Application: Learning Halfspaces with Massart Noise

In this section, we show how to apply the machinery of Forster decompositions from Section 2 to PAC learn halfspaces with Massart noise. Specifically, we will show how to adapt the algorithm of [DGT19], by appropriately transforming the set of points it is run on, to obtain a new algorithm with strongly polynomial sample complexity guarantees.

To that end, we will need the definition of an outlier and a partial classifier:

**Definition 3.1.** ($\Gamma$-Outlier) A point $x$ in the support of a distribution $X$ over a vector-space $V$ is called a $\Gamma$-*outlier*, $\Gamma > 0$, if there exists a vector $v \in V$ such that $|v \cdot x| > \Gamma\sqrt{\mathbf{E}[|v \cdot X|^2]}$.

**Definition 3.2.** (Partial Classifier) A *partial classifier* is a function $h : \mathbb{R}^d \to \{-1, *, 1\}$. It can be thought of as acting as a classifier that for some input values returns an output in $\{\pm 1\}$, and for the remaining values returns $*$, as a way of saying "I don't know".

A key step of the algorithm in [DGT19] for learning halfspaces with Massart noise, is computing a partial classifier that returns an output on a non-trivial fraction of inputs for which its error rate is at most $\eta + \epsilon$. The sample complexity and running-time of this algorithm depends polynomially on the size of the largest outlier in the distribution and the inverse of the accuracy parameter $1/\epsilon$. More specifically, the following theorem is implicit in the work of [DGT19].

**Theorem 3.3** ([DGT19]). *Let $V$ be a vector space and $(X, Y)$ a distribution over $V \times \{\pm 1\}$, where $X$ does not have any $\Gamma$-outliers in its support. Suppose that there is a vector $w$ and $\eta \in (0, 1/2)$ such that for any given value $x$ it holds $\mathbf{Pr}[Y = \mathrm{sign}(w \cdot x) \mid X = x] \geq 1 - \eta$. In particular, $Y$ is given by a homogeneous halfspace with at most $\eta$ Massart noise. Then there exists an algorithm that, given $\eta, \Gamma$, and parameters $\epsilon', \delta' > 0$, draws $\mathrm{poly}(\dim(V), \Gamma, 1/\epsilon', \log(1/\delta'))$ samples from $(X, Y)$, runs in sample-polynomial time, and returns a partial classifier $h$ on $V$ such that with probability at least $1 - \delta'$ the following holds: (i) $\mathbf{Pr}[h(X) \neq *] > 1/\mathrm{poly}(\dim(V), \Gamma, 1/\epsilon', \log(1/\delta'))$, and (ii) $\mathbf{Pr}[h(X) \neq Y \mid h(X) \neq *] < \eta + \epsilon'$.*

The main idea behind obtaining an efficient algorithm with strongly polynomial sample complexity is to repeatedly apply the Forster decomposition theorem from Section 2 to ensure that no large outliers exist on a "heavy" subspace. Then, using Theorem 3.3, we can identify a partial classifier that has small misclassification error on a non-trivial fraction of the probability mass for which it outputs a $\{\pm 1\}$ prediction. By recursing on the remaining probability mass, we can ensure that we accurately predict the label for nearly all the distribution of points. This yields a misclassification error of at most $\eta + \epsilon$ in the entire space.

The following theorem summarizes the main algorithmic learning result of this paper.

**Theorem 3.4** (Main Learning Result). *Let $(X, Y)$ a distribution over $\mathbb{R}^d \times \{\pm 1\}$, where $Y$ is given by a homogeneous halfspace in $X$ with at most $\eta < 1/2$ rate of Massart noise, and where the elements in the support of $X$ are all integers with bit complexity at most $b$. For parameters $\epsilon, \delta > 0$, there exists an algorithm that draws $\mathrm{poly}(d, 1/\epsilon, \log(1/\delta))$ samples from $(X, Y)$, runs in $\mathrm{poly}(d, b, 1/\epsilon, \log(1/\delta))$ time, and with probability at least $1 - \delta$ returns a classifier $h : \mathbb{R}^d \to \{\pm 1\}$ with misclassification error at most $\eta + \epsilon$.*

*Proof.* In order to analyze Algorithm 1, we will say that an event happens with "high probability" if it happens with probability at least $1 - \delta/\mathrm{poly}(d \log(1/\delta)/\epsilon)$ for a sufficiently high degree polynomial. There are a number of events in each iteration of our while loop that we will want to show happen with high probability, and we will later claim that if they do for each iteration of the loop, then our algorithm will return an appropriate answer after $\mathrm{poly}(d \log(1/\delta)/\epsilon)$ iterations of the while loop. This will imply that with probability at least $1 - \delta$ all high probability events occur and that our algorithm will return an appropriate answer.

We start by noting that with high probability the sample set chosen in the check of the while statement approximates the true probability that $h_i(X) = *$ to additive error at most $\epsilon/6$. This means that, with high probability, (1) we will not break out of the while loop unless this probability is less than $\epsilon/2$, and that (2) while we are in the while loop, $h(X) = *$ with probability at least $\epsilon/6$. This latter statement implies that the expected number of samples from $(X, Y)$ needed in order to find one with $h(X) = *$ is $O(1/\epsilon)$. Assuming this holds, the set $S$ in the next line can be found with high probability by taking a polynomial number of samples from the original distribution.

Line 6 runs in deterministic $\mathrm{poly}(db \log(1/\delta)/\epsilon)$ time and, by Proposition 2.4, with high probability finds a pair $V, A$ such that:

1. $\mathbf{Pr}[X \in V : h_i(X) = *] \geq 1/(2d)$.

2. $\mathbf{E}[f_A(X)f_A(X)^T : X \in V, h_i(X) = *] > (1/(4\dim(V))) I_V$.

If Condition 1 holds, then with high probability only polynomially many samples from $(X, Y)$ are needed to run the algorithm in Line 7. If Condition 2 holds, then the conditional distribution has no $(4\dim(V))$-outliers. Since $(f_A(X), Y)$ is a linear classifier with Massart noise $\eta$, with high probability we have that $\mathbf{Pr}[g(f_A(X)) \neq * \mid X \in V, h_i(X) \neq *] > 1/\mathrm{poly}(d \log(1/\delta)/\epsilon)$ and

$$\mathbf{Pr}\left[g(f_A(X)) \neq Y \mid h_i(X) = *, X \in V, g(f_A(X)) \neq *\right] < \eta + \epsilon/2 .$$

The former statement implies along with Condition 1 that

$$\mathbf{Pr}[h_{i+1} \neq * \mid h_i(X) = *] > 1/\mathrm{poly}(d\log(1/\delta)/\epsilon) ,$$

and the latter implies that

$$\mathbf{Pr}[h_{i+1}(X) \neq Y \mid h_{i+1}(X) \neq *] \leq \max(\eta + \epsilon/2, \mathbf{Pr}(h_i(X) \neq Y | h_i(X) \neq *)) .$$

If the above hold for every iteration of the while loop, then after $T = \mathrm{poly}(d\log(1/\delta)/\epsilon)$ iterations, we will have that $\mathbf{Pr}[h_T(X) = *] < \epsilon/6$ (and thus with high probability we will break out of the loop in the next iteration), and when we do break out, it holds that

$$\mathbf{Pr}[h_i(X) \neq Y] \leq \mathbf{Pr}[h_i(X) \neq Y \mid h_i(X) \neq *] + \mathbf{Pr}[h_i(X) = *] \leq (\eta + \epsilon/2) + (\epsilon/2) = \eta + \epsilon .$$

This completes the proof of Theorem 3.4. $\qquad\square$

---

**Algorithm 1** Learning Algorithm for Massart Halfspaces
---
1: Let $C > 0$ be a sufficiently large universal constant.
2: Let $h_0 : \mathbb{R}^d \to \{-1, *, 1\}$ always return $*$.
3: Let $i = 0$
4: **while** $\mathbf{E}_{x \sim_u \hat{S}}[h_i(x) = *] > \epsilon/3$ for a random sample $\hat{S}$ of $C \log(d\epsilon/\delta)/\epsilon^2$ points $(X, Y)$ **do**
5: $\quad$ Let $S$ be a set of $Cd^4 \log(1/\delta)$ samples from $X$ conditioned on $h_i(X) = *$.
6: $\quad$ Run the algorithm from Theorem 1.4 on $S$ to find a subspace $V$ and linear transformation $A : V \to V$ with $\mathbf{E}_{x \sim_u S \cap V}[f_A(x)f_A(x)^T] > (1/(2\dim(V))) I_V$.
7: $\quad$ Obtain a partial classifier $g$ by running the algorithm from Theorem 3.3 with:

$$\delta' = \delta/\mathrm{poly}(d\log(1/\delta)/\epsilon), \quad \epsilon' = \epsilon/2, \quad \text{and} \quad \Gamma = 4\dim(V)$$

$\quad$ on the distribution $(f_A(X), Y)$ over all $(X, Y)$ conditioned on $X \in V$ and $h_i(X) = *$.
8: $\quad$ Define a new partial classifier

$$h_{i+1}(x) = \begin{cases} g(f_A(x)) & \text{if } h_i(x) = * \text{ and } x \in V \\ h_i(x) & \text{otherwise} \end{cases}$$

9: $\quad$ Set $i \leftarrow i + 1$.
10: Return the classifier $h_i$.

---

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
