# APPENDIX

## A  Omitted Details from Proposition 2.2

Relaxing the constraints of the SDP by $\delta$ guarantees that if the original SDP has a solution $\{c^2(x)\}$, then the new SDP will have a solution set containing a box of volume at least $(\delta/\dim(V))^d$ defined with variables $c_0^2(x)$ satisfying $c^2(x) \le c_0^2(x) \le (1 + \delta/\dim(V))c^2(x)$. It is easy to see that these solutions satisfy the necessary constraints. In order to show that the ellipsoid algorithm will work, it will suffice to show that this box can be taken to be contained in a ball of radius $R$. This will imply that the ellipsoid algorithm will find a solution to the relaxed SDP assuming one existed for the original in time $\mathrm{poly}(d, \log(R/\delta))$. We will in fact show, using a more refined version of Forster's theorem from [AKS20], that $R$ can be taken to be $n^{\mathrm{poly}(b,d)}$.

We leverage (i) the constraint that no subspace of dimension $\kappa$ contains more than $\kappa \cdot \dim(V)/n$ points and (ii) that all coordinates are integers bounded by $2^b$, to argue that the function $c^2$ must take values within a bounded range. We will first prove this for the case where $\dim(V) = d$, and argue that the same bound holds for the general case.

Towards this end, we will use Theorem 1.5 from [AKS20], which states that if one has a collection of unit-norm points which are in "$(\eta, \delta)$-deep position", they can be brought in radial isotropic position by rescaling points with factors between 1 and $n/(\eta\delta)^{O(d)}$.

For a (unit-norm) point-set $X$ to be $(\eta, \delta)$-deep according to the standard radial isotropic transformation, they require that for any subspace $E^\kappa$ of dimension $\kappa$, the number of points lying within Euclidean distance $\delta$ from that subspace is at most $(1 - \eta)\kappa n/d$, i.e., for the set $E_\delta^\kappa = \{x \in X : d(x, E^\kappa) \le \delta\}$ it holds $|E_\delta^\kappa| \le (1 - \eta)\kappa n/d$.

We now show that the condition is satisfied for $\eta = 1/(nd)$ and $\delta = \frac{1}{2d}2^{-b}d^{-d}$. Since for any set $S$ of $d$ linearly-independent points with integer coordinates, the determinant $|X_S|$ is at least 1, after renormalizing so that all points are unit norm, the determinant is at least $\Delta = 2^{-bd}d^{-d}$. As it shown in Lemma 4.6 of [AKS20], choosing $\delta = \sqrt{\Delta}/(2d)$ ensures that the set $E_\delta^\kappa$ lies in a $\kappa$-dimensional subspace. Moreover, since for any set $S$ of points lying in a $\kappa$-dimensional subspace for $\kappa < \dim(V)$, it holds that $|S| < \kappa n/\dim(V)$, this implies that $|S| \le (1 - 1/(nd))\kappa n/\dim(V)$. Thus, for our choice of $\kappa$ and $\delta$, the given point-set is $(\eta, \delta)$-deep.

The same argument goes through if the points lie on a subspace of dimension $\dim(V) < d$. The only subtle point is bounding the $\dim(V)$-dimensional volume of any parallelepiped defined by $\dim(V)$ linearly independent points. This corresponded to the absolute value of the determinant when the point-set was full dimensional. This volume is given by $\sqrt{|\det X_S^T X_S|}$, where $X_S$ is the matrix with the points in $S$ written as columns. For any point-set of integer coordinates this determinant is at least 1. After renormalizing all the points so that they have unit-norm, this determinant is at least $2^{-db}d^{-d}$, as before.

Overall, we get that the renormalizing factors $c^2$ are between 1 and $n^{\mathrm{poly}(b,d)}$.