# OpenReview forum: "Forster Decomposition and Learning Halfspaces with Noise"
_NeurIPS.cc/2021/Conference — NeurIPS 2021 Spotlight_

### Official Review · Reviewer_Nd9v · 2021-07-02

**Rating:** 7
**Confidence:** 4

**Summary:**

The paper studies learning halfspaces with Massart noise. The main contribution is a new data preprocessing scheme that results in bit-independent sample complexity.

**Limitations And Societal Impact:**

yes

**Main Review:**

+ Authors clearly state the root of bit dependence in prior works which well motivates a new data preprocessing scheme.

+ The intuition underlying the approach is clearly explained. In particular, the problem boils down to finding a linear transformation of the data such that each data point has unit norm and the covariance matrix is nearly identity.

Below are my concerns.

- It was argued that Theorem 1.4 is more general than [AKS20]. Can you clarify the difference and why such algorithmic extension is crucial for the problem of interest?

- Section 2 is hard to follow, especially for those not familiar with [For02].

- Why does the running time still depend on the bit complexity?

- Are there other applications of the improved Forster decomposition beyond learning halfspaces?

**Time Spent Reviewing:**

5

---

> ### Author Response · Authors · 2021-08-10
> **Author response**
>
> We thank the reviewer for their positive comments. We address the reviewer's questions below.
>
>
> We note that [AKS20] provides an algorithm to compute a Forster transform *when one exists*, but gives no guarantees when it doesn’t.
> In situations where a large fraction of the points lie in a lower dimensional subspace, a Forster transform does not exist
> but learning is still possible. Theorem 1.4 allows us to identify the lower-dimensional subspace and to recursively solve the problem
> on a restricted point-set. This is crucial for the problem of PAC learning halfspaces, where one does not have any guarantees
> about the points that the algorithm receives.
>
> In terms of the runtime dependence on the bit complexity, this comes in because of the need to solve an SDP to compute the Forster transform. In particular, for our application of learning halfspaces, even without any noise, all known algorithms have a polynomial dependence on the bit-complexity of the points in their runtime. In fact, finding a linear separator in the noiseless setting
> is equivalent to solving linear programs; it is a major open question in computer science whether linear programs are solvable
> without a polynomial dependence on the bit complexity of the input, i.e., obtaining a strongly-polynomial algorithm for LPs.
>
> Finally, we note that our technique of Forster Decompositions and our algorithmic results can be useful in a variety
> of other contexts.  They imply improvements to the problem of subspace recovery and can be useful
> as a preconditioning step in other learning tasks, where one can rotate points and scale them without changing the task.

---

> > ### Comment · Reviewer_Nd9v · 2021-08-11
> > **reviewer comments**
> >
> > Thank you for the response. It addressed all my conerns.

---

### Official Review · Reviewer_px8j · 2021-07-13

**Rating:** 6
**Confidence:** 3

**Summary:**

The paper presents an algorithm for halfspace-learning with data supported on points with integer coordinates of bounded bit complexity under partially adversarial noise conditions. A related result in the literature is improved insofar as the sample complexity of the algorithm is shown to be independent of the bit complexity of the support.
The principal technical contribution is an iterative method of deconcentration of the data.

**Ethical Concerns:**

There are no ethical issues

**Limitations And Societal Impact:**

Adequately addressed

**Main Review:**

The paper contains some interesting ideas, but I found it very hard to understand, partly because I am not an expert specializing on this subject, but mainly because the presentation is very unclear.

The assumption that the input marginal is concentrated on points with integral coordinates is hidden from the reader until page 4, and up to this point one remains bewildered by the frequent use of the term "bit-complexity".

The paper leans very heavily on the reference DGT19, without consultation of which it  would be nearly impossible to understand.

The same applies to the reference HKLM20, cited in Theorem 2.1. As given the citation is meaningless, because the transformation A, whose existence is asserted, does not appear in the conclusion. This may be a simple typo, but it makes it impossible to follow the proof of Proposition 2.2. After consultation of HKLM20 Proposition 2.2 becomes trivial.

Halfspaces are denoted h on page one and f in Theorem 1.2, but then f is also used in the definition of the normalized transformations in Definition 1.3.

The proposed method depends crucially on the assumption that the halfspaces are homogeneous. In line 79 this is assumed "without loss of generality", "since we are working in the distribution-independent setting". Could this be explained a little more? How do you find a point on the separating hyperplane to be used as an origin?

Another important ingredient of the final algorithm is Theorem 3.3, attributed to DGT19. Where can we find it in DGT19?

The page limit is not to blame for the lack of clarity in the paper. The introduction could be a lot less verbose to make ample room for necessary definitions and explanations.

POST REBUTTAL
----------------------

The authors addressed most of my concerns in a clear and satisfactory way, and I will raise my score accordingly.
Nevertheless I still believe that the presentation should be more transparent.

**Time Spent Reviewing:**

8

---

> ### Author Response · Authors · 2021-08-10
> **Author response**
>
> We thank the reviewer for their time and effort in providing feedback.
> We address concrete points made by the reviewer below.
>
> * Bit-complexity: The definition of the bit-complexity of a point is standard in the related literature on this and related problems.
>
> * Confusion on Theorem 2.1/Proposition 2.2 statement: Indeed, this is a typo. In the conclusion of Theorem 2.1,
>   Sigma_V should actually be Sigma_A, which is defined as E_{x in V}[f_A(x)f_A(x)^T]. We thank the reviewer for catching this.
>
> * Homogeneous halfspaces: One can reduce the problem of learning arbitrary halfspaces to the problem of learning
> homogenous halfspaces by adding an extra constant coordinate to every sample, i.e., the point (x, y) becomes ((x, -1), y).
>
> * Theorem 3.3 is not explicitly stated in DGT19, but follows directly from their work (specifically, Algorithm 2
>  and the proof of Theorem 2.9 in the conference version).

---

> > ### Comment · Reviewer_px8j · 2021-08-11
> > **Thank you**
> >
> > Thank you. You addressed all my concerns in a satisfactory way. With respect to homogeneous halfspaces you probably wanted to write (x,y) becomes ((x,y),y) - something I should have thought of myself.
> > I will raise my score.

---

### Official Review · Reviewer_91N9 · 2021-07-15

**Rating:** 8
**Confidence:** 4

**Summary:**

This paper provides an algorithm that can efficiently decompose a distribution into a disjoint mixture of a few distributions for which Forster transforms exist and can be computed efficiently. They then use this to obtain the first polynomial-time algorithm for distribution independent PAC learning of halfspaces in the Massart noise model. This is the first algorithm to actually obtain a result that scales independent of the bit complexity of the examples.

The authors demonstrate that it is either possible to find a transformation c(x) such that while the distribution on the x's itself might not be transformable via a Forster transform, c(x) x is; or to identify a subspace W of V (the space in which the x's exist) which contains more than a dim(W)/dim(V) fraction of the samples. In the latter case, one can then focus on the subspace and then rescale appropriately within the subspace.

To solve the problem of learning halfspaces with Massart noise, the authors repeatedly apply the Forster decomposition theorem to ensure that no large outliers exist on a heavy subspace, and train partial classifiers at each step of the algorithmic transformation.


**Limitations And Societal Impact:**

This is a theoretical result and of limited societal impact.

**Main Review:**

I find this result very interesting. I think it is clear, original as well as significant. I think the algorithmic version of the result from [HKLM20] is a very nice contribution.

I have read the author response and I did not have any questions for the authors.

**Time Spent Reviewing:**

3

---

> ### Author Response · Authors · 2021-08-10
> **Author response**
>
> We would like to thank the reviewer for appreciating our work.

---

### Official Review · Reviewer_48Ey · 2021-07-16

**Rating:** 8
**Confidence:** 4

**Summary:**

This paper addresses the problem of distribution-independent improper learning of halfspaces under the Massart noise model. The prior work [DGT19], in a breakthrough, provided the first algorithm with time and samples $\text{poly}(d, 1/\epsilon, b)$ where $d$ is dimension, $\epsilon$ is excess error (compared to the Massart noise bound $\eta$), and $b$ is the bit complexity of the samples. The dependence of runtime on the bit complexity $b$ is necessary, barring a breakthrough in solving LPs, but the sample dependence on $b$ is information-theoretically unnecessary, leaving a statistical/computational gap. The main result of this paper is to close this gap, providing an algorithm with same time complexity as before, but sample complexity only $\text{poly}(d, 1/\epsilon)$.

The innovation of the current work which enables this result is an efficient algorithm for decomposing a set of points into subsets each of which has a Forster transform (a linear transformation such that if the points are rescaled to the unit sphere, the covariance is lower bounded by $1/\text{poly}(d))$. Recent work [HKLM20] showed that every set of points either has lots of points lying in a subspace, or has a Forster transform. This work uses that result to provide polynomial-time algorithms both for finding a heavy subspace (if it exists) and finding the Forster transform (otherwise); applying these recursively yields the decomposition. Finally, the halfspace learning result follows by applying an algorithm from [DGT19] for learning halfspaces when the data has a large margin (more precisely, [DGT19] provides a partial learning algorithm, and the main algorithm recurses on the points which the partial learner doesn't label).

**Limitations And Societal Impact:**

Yes.

**Main Review:**

This work provides an improvement on the long-standing problem of learning halfspaces with semi-adversarial noise. The main idea is a simple synthesis of the results of [DGT19] and [HKLM20], but the execution is non-trivial; the authors claim that prior algorithms for finding Forster transforms and heavy subspaces do not apply in the needed generality. The technique of recursive Forster decomposition may have other applications in learning theory.

The paper is very clearly written and a pleasure to read.

NOTES
- Why do the results of [AKS20] and [HM13] not apply in the needed generality? Explaining this would clarify a main part of the novelty of the paper.
- I appreciate the intuition about the Forster transform but I'm not sure why lines 208-221 are helpful. It seems to me that lines 222-223 are the crucial part and the rest is making it seem more complicated than it needs to be?

**Time Spent Reviewing:**

5

---

> ### Author Response · Authors · 2021-08-10
> **Author Response**
>
> We thank the reviewer for the appreciation of our paper and their positive comments.
> We address specific questions by the reviewer below.
>
> We start by pointing out that the main contribution of our work is conceptual. The high-level message of our work is
> that Forster decompositions provide the "correct" pre-processing procedure to eliminate the bit complexity
> from the sample size for the problem of efficiently learning halfspaces with Massart noise. In more detail:
> - We relax the notion of the Forster transform to a Forster decomposition, so that it applies to any point-set,
> and we provide an efficient algorithm to compute such a decomposition; and
> - We provide a novel application of this technique of Forster decompositions in a well-studied learning setting,
> giving the first computationally efficient algorithm with strongly-polynomial sample complexity
> for learning halfspaces in the presence of Massart noise.
>
> The prior results of [AKS20] and [HM13] do not provide the desirable guarantees when a Forster transform does not exist,
> which is crucial for our learning application. In particular, our Proposition 2.3 is new and not covered by these prior works.
>
> On the other hand, Proposition 2.2 is not a major contribution of our work, and its (short) proof is given mainly
> for the sake of the presentation. In particular, the algorithm of [AKS20] or [HM13] could have been used
> in place of our SDP formulation in Proposition 2.2 for computing Forster transforms when no heavy subspace exists.
> (We will clarify this point in the final version of the paper.)
> However, the application of these results in our setting would require additional work to establish
> that the required conditions for computing such a transform are satisfied by point-sets with bounded bit complexity.
> As a side note, we point out the analysis of [HM13] actually has a bug, pointed out and corrected in [AKS20].
> To the best of our understanding, the analysis in [AKS20] is correct and their main theorem could have been used
> in place of our Proposition 2.2, modulo the aforementioned technicalities. For the sake of readability, we wanted
> to provide a self-contained version of the algorithm itself. Moreover, we believe that the direct SDP formulation given in our
> Proposition 2.2, which is different from the approach taken in the aforementioned prior works, is more succinct
> and intuitive and facilitates the presentation of our results.
>
> Lines 208-221 are intended to provide intuition which motivates (and natural leads to) the definition of a Forster transform/decomposition.

---

> > ### Comment · Reviewer_48Ey · 2021-08-13
> > **Thanks**
> >
> > Thank you for the response; it answers my questions.

---

### Decision · Program_Chairs · 2021-09-28

**Decision:**

Accept (Spotlight)

**Comment:**

This paper makes a foundational advance following-up on the recent breakthrough on learning halfspaces with Massart noise. The algorithms of Diakonikolas, Gouleakis, Tzamos from 2019 learns halfspaces with Massart noise but due to an LP solving step (and the absence of a strongly polynomial algorithm for LP), runs in time and samples that grows polynomially in the bit complexity of the coefficients of the underlying halfspace. A polynomial dependence on bit complexity in running time is reasonable but one could hope for an algorithm where the sample complexity is independent of it.

This paper cleverly uses (an algorithmic version) of Forster decomposition of points in order to resolve this question. The idea, in retrospect, is simple but elegant and (the algorithmic variant of Forster decomposition can potentially find new applications. We recommend acceptance.

**Consistency Experiment:**

NeurIPS has a long history of experimentation. In 2014, NeurIPS ran an experiment in which 10% of submissions were reviewed by two independent committees to quantify the randomness in the review process. This year, we repeated a variant of this experiment to see how the quality of the review process has changed over time.  This paper was part of the experiment and was therefore assigned to two committees (consisting of reviewers, an Area Chair, and a Senior Area Chair) that reached independent decisions.  If both committees made the same recommendation, this recommendation was followed. If a single committee recommended acceptance, the paper was accepted (with the exception of a few cases in which the other committee identified what we considered a fatal flaw, e.g., an error in a key result).

This copy’s committee reached the following decision: **Accept (Spotlight)**

The other committee assigned to the paper recommended **Accept (Poster)**.  You can find the other set of reviews, along with any follow up discussion with the authors here:
https://openreview.net/forum?id=l4DQWgjbZg